# *InArt*: In-Network Aggregation with Route Selection for Accelerating Distributed Training

## ABSTRACT

Deep learning has brought about a revolutionary transformation in network applications, particularly in domains like e-commerce and online advertising. Distributed training (DT), as a critical means to expedite model training, has progressively emerged as a key foundational infrastructure for such applications. However, with the rapid advancement of hardware accelerators, the performance bottleneck in DT has shifted from computation to communication. In-network aggregation (INA) solutions have shown promise in alleviating the communication bottleneck. Regrettably, current INA solutions primarily focus on improving efficiency under the traditional PS architecture and do not fully address the communication bottleneck caused by limited PS ingress bandwidth. To bridge this gap, we propose InArt, the first work to introduce INA with routing selection in a multi-PS architecture. InArt employs a multi-PS architecture to split DT tasks among multiple PSs and selects appropriate routing schemes to fully harness INA capabilities. To accommodate traffic dynamics, InArt adopts a two-phase approach: splitting the training model among multiple parameter servers and selecting routing paths for INA. We propose Lagrange multiplier and randomized rounding algorithms for these phases, respectively. We implement InArt and evaluate its performance through experiments on physical platforms (Tofino switches) and mininet emulation (P4 Software Switches). Experimental results show that InArt can reduce communication time by 49% compared with state-of-the-art solutions.

## CCS CONCEPTS

• **Networks → In-network processing**; • **Computing methodologies → Machine learning**.

**ACM Reference Format:**
Anonymous Author(s). 2024. *InArt*: In-Network Aggregation with Route Selection for Accelerating Distributed Training. In *The Web Conference 2024 (WWW '24), May 13-17, 2024, Singapore*. 10 pages. https://doi.org/xxxxxx

## 1 INTRODUCTION

Over the past decade, deep learning has become an essential component of many web applications [1–4]. It plays a crucial role in domains such as e-commerce [2], social media [3], and online advertising [4], enabling the development of personalized recommendation systems, content analysis, and targeted advertising. The success of deep learning solutions lies in their sophisticated models, which contain numerous parameters and are trained on substantial amounts of data [5, 6]. However, training such models is time-consuming and computationally demanding. For instance, training a BERT model with 110 million parameters on a single server takes over 1.5 months [7]. To address this bottleneck and expedite model training, the adoption of DT is widespread in web infrastructure [5, 6, 8]. By harnessing the power of DT, web applications can efficiently process large datasets and leverage advanced deep learning models to deliver high-quality predictions and decision-making.

Following the typical parameter server (PS) architecture [9, 10], a DT system usually consists of a PS and multiple workers that perform many rounds of iterative training. In each iteration, workers compute local gradients and send gradients to the PS for aggregation. The above two processes are called *gradient computation* and *gradient aggregation*, respectively [11]. On the one hand, the rapid development of hardware accelerators (*e.g.*, GPU and FPGA) can significantly improve computing speed. On the other hand, considering that deep learning models employed in web applications often possess a substantial number of parameters, (*e.g.*, BERT [7] with about 110 million parameters, which is widely used in online advertising), the DT will introduce several gigabytes of data transfers. As a result, the performance bottleneck of DT has *shifted from computation to communication* [5, 11, 12]. For example, for training the DT job of BERT on 10Gbps links, more than half of the DT time is spent on communication [5].

To alleviate the communication bottleneck in DT, previous works usually focus on gradient compression [14–16] or communication scheduling [17–19]. However, gradient compression will inevitably lead to training accuracy degradation, and communication scheduling does not reduce the traffic volume and may still encounter the communication bottleneck on links or/and PSs. With the advent of programmable network hardware (*e.g.*, programmable switches [20] and smartNICs [21]), *in-network aggregation (INA)* [5, 11, 22] holds great promise in solving the communication bottleneck. Specifically, some gradients can be aggregated by programmable devices inside the network. In this way, the traffic volume sent to the PS will be reduced, thereby alleviating the inbound bandwidth bottleneck of the PS (see details in §2.1).

Accelerating DT with INA is complicated, and only a few works [5, 11] have made preliminary exploration in this field. SwitchML [5] aggregates gradients on top-of-rack (ToR) programmable switches of workers to minimize communication overheads at a single-rack scale. ATP [11] proposes a protocol based on P4-programmable switches to support INA for multi-tenant learning across racks. GRID [23] addresses the selection of appropriate gradient aggregation points for each worker in a DT cluster. However, these works primarily focus on improving INA effectiveness under the traditional PS architecture. In reality, under this architecture, as training tasks scale up, the PS's ingress bandwidth may not meet

**Figure 1: A DT system consists of two PSs (*i.e.*, $S_1$ and $S_2$), six workers (*i.e.*, $W_1$-$W_6$) and two programmable switches (*i.e.*, $V_1$ and $V_2$). The switch processing capacity, PS processing capacity and link bandwidth are set to 6, 4 and 6, respectively. We use colors to distinguish gradients sent to different PSs. Values near PSs and switches represent their loads, while those near workers denote maximum gradient sending rates. The left plot shows the gradient communication scheme using the LBMM method [13], and the maximum sending rate is $4/3$. The middle plot shows the gradient communication scheme by ATP [11] with a maximum sending rate of $2$. The right plot shows that our proposed InArt can achieve a maximum sending rate of $2.5$.**

the demands of synchronizing parameters, especially in DT clusters with numerous worker nodes. The INA scheme alone cannot fully resolve the communication bottleneck at the PS due to the transmission of a significant volume of gradient updates.

To tackle this challenge, we introduce a multi-PS architecture [24–26] within the INA scheme. Notably, the multi-PS architecture and INA are two complementary and mutually beneficial approaches for alleviating communication bottlenecks in DT. By employing a multi-PS architecture to split DT tasks among multiple PSs and selecting appropriate routing schemes to fully harness the INA capabilities, we can effectively alleviate communication bottlenecks and enhance the efficiency of DT (as demonstrated in §5). Therefore, *there is an urgent need for INA solutions with route selection in multi-PS architectures*. However, it is a non-trivial mission to achieve the goal. Firstly, considering multiple PSs in a cluster for gradient aggregation, we need to decide which PS each gradient should be sent to, *i.e.*, *the routing destination is uncertain*. Secondly, aggregating gradients on programmable switches will change the traffic size in forwarding, *i.e.*, *the routing traffic volume is variable*. Thirdly, performing DT will face multi-dimensional *resource constraints*, such as switch/PS processing capacity and link bandwidth. To address these challenges, this paper proposes InArt and the main contributions are as follows:

- We design InArt, the first-of-its-kind INA work with routing selection in a multi-PS architecture for accelerating DT.
- Due to traffic dynamics, we take a two-phase approach: splitting the training model among multiple PSs and selecting the routing paths for INA. For the first phase, we propose a Lagrange multiplier-based algorithm, called L-InArt. For the second phase, we design a randomized rounding-based algorithm, named R-InArt.
- We implement InArt on both the hardware testbed (with two Tofino hardware switches) and software emulation (with bmv2 software P4 switches). Experimental results show that InArt achieves superior performance compared with state-of-the-art solutions.

## 2 MOTIVATION

### 2.1 A Motivation Example

This section illustrates the pros and cons of state-of-the-art solutions through an example, which motivates our study. As shown in Fig. 1, a DT system using a multi-PS architecture consists of two PSs (*i.e.*, $S_1$ and $S_2$), six workers (*i.e.*, $W_1$-$W_6$) and two programmable switches (*i.e.*, $V_1$ and $V_2$). For simplicity, the switch processing capacity, PS processing capacity and link bandwidth are set to 6, 4 and 6, respectively, and the unit is omitted. In the example, we first need to split the model, and each PS maintains a certain partition of the model, *i.e.*, *sub-model*. Then, we need to decide at what rate the gradients should be sent to the corresponding PS under various resource constraints. In this paper, we consider the typical *synchronous* scheme, *i.e.*, PSs will aggregate the gradients after receiving gradients from all required workers [8]. Usually, a faster gradient sending rate means a shorter communication time. Thus, our objective is to maximize the gradient sending rate of workers.

Let's first consider the Load Balance Min-Min (LBMM) algorithm [13], a classical algorithm used in multi-PS architecture without INA. LBMM selects the link with the lightest load for load balancing routing. As shown in Fig. 1(a), each worker sends half of the gradients to $S_1$ for aggregation and the other half to $S_2$ for aggregation. In addition, since the total processing capacity of the two PSs is 8 and there are six workers, the maximum gradient sending rate of each worker should be $4/3$ (*i.e.*, $2/3$ to $S_1$ and $2/3$ to $S_2$). Otherwise, PSs will be overloaded.

We then consider a recent work on INA, called ATP [5]. Specifically, ATP performs best-effort aggregation on the ToR switch for each worker under the corresponding rack, and the results are shown in Fig. 1(b). Since ATP does not involve the splitting of the model, here we assume the model is split equally, with each worker sending half of the gradients to $S_1$ and the other half to $S_2$. In this case, gradients from $W_1$, $W_2$ to $S_1$ are aggregated on switch $V_1$, gradients from $W_3$, $W_4$ to $S_1$ are directly routed to $S_1$ without INA, gradients from $W_1$, $W_2$, $W_3$, $W_4$ to $S_2$ are aggregated on switch

$V_1$, gradients from $W_5$, $W_6$ to $S_1$ and $S2$ are aggregated on switch $V_2$. Therefore, the maximum gradient sending rate of ATP is 2.

## 2.2 Our Intuition

A question immediately following the above discussion is *can we do better by combining the merits of LBMM and ATP?* In Fig. 1(c), we demonstrate that by selecting an appropriate partitioning scheme among multiple PSs and implementing an optimal routing scheme for INA, we can achieve a maximum gradient sending rate of 2.5. This rate is 87.5% faster than LBMM and 25% faster than ATP. In this case, gradients from $W_1$, $W_2$, $W_3$, $W_4$ to $S_1$ are aggregated on switch $V_1$, gradients from $W_5$ and $W_6$ to $S_1$ are aggregated on switch $V_2$, gradients from $W_1$, $W_2$, $W_3$ to $S_2$ are directly routed to $S_2$ without INA and gradients from $W_4$, $W_5$, $W_6$ to $S_2$ are aggregated on $S_2$. With our findings, *this paper aims to accelerate distributed training by designing an efficient route selection in a multi-PS architecture for INA, with the objective of maximizing the gradient sending rate.*

## 3 PROBLEM DEFINITION

### 3.1 System Model

A typical multi-PS architecture mainly contains two components, worker set $W = \{w_1, ..., w_{|W|}\}$ and PS set $S = \{s_1, ..., s_{|S|}\}$, where $|W|$ and $|S|$ are the numbers of workers and PSs, respectively. According to the above definition, we model the DT cluster as $G = (W, S, V, E)$, where $V = \{v_1, ..., v_{|V|}\}$ is the set of programming switches (*e.g.*, Intel Tofino switches [20]), and $E = \{e_1, ..., e_{|E|}\}$ represents the communication links among these switches, workers and PSs. During the process of gradient aggregation, we regard gradients with the same source (*i.e.*,worker) and aggregation location (*i.e.*, switch or PS) as a *flow* for simplicity. Let $P_{s,w}$ denote a set of feasible routing paths from worker $w$ to PS $s$. Similarly, let $P_{v,w}$ and $P_{s,v}$ denote the feasible routing path set from switch $v$ to worker $w$ and from PS $s$ to switch $v$, respectively. Moreover, we use $P = \{P_{s,w} \cup P_{v,w} \cup P_{s,v} \mid \forall w \in W, v \in V, s \in S\}$ to denote the all feasible routing path in the cluster $G$.

For each switch $v \in V$, we use $C(v)$ to denote the total processing capacity, and $c(v)$ to denote the processing capacity used by background traffic. Moreover, for each PS $s \in S$, let $B(s)$ represent the total ingress bandwidth, and $b(s)$ represent the amount of ingress bandwidth already occupied. Similarly, let $B(e)$ and $b(e)$ denote the total bandwidth and the bandwidth used by background traffic for each link $e$, respectively. For simplicity, we focus on accelerating the training time of a single DT job in this paper. Actually, for a single DT training job, the traffic of other jobs can be considered as background traffic since different DT jobs are independent of each other [11, 27]. Therefore, the proposed scheme can be easily extended to multi-DT job scenarios.

### 3.2 Problem Definition of InArt

The key idea of InArt is to make the following three decisions.

- The proportion of the model aggregation that each PS is responsible for. Let variable $x_s$ represent the proportion of the model that the PS $s$ is responsible for aggregation.
- The location where each gradient is aggregated. Let variable $y^{s,w} \in \{0, 1\}$ denote whether gradients from worker $w$ to

PS $s$ are directly aggregated on the PS $s$ or not. We use variable $y_v^{s,w} \in \{0, 1\}$ to represent whether gradients from worker $w$ to PS $s$ are aggregated by programmable switch $v$ or not.

- The routing path for each gradient. Let binary variables $q_p^{s,w}$, $q_p^{v,w}$ and $q_p^{s,v}$ denote whether gradients from worker $w$ to PS $s$, from worker $w$ to switch $v$ and from switch $v$ to PS $s$ will be routed on path $p$ or not, respectively.

We further consider the following six constraints when performing INA with route selection.

- *Model partition constraints:* We split the model into several sub-models and each PS is responsible for a sub-model. This means that each sub-model must have a corresponding PS for aggregation, represented as the equation $\sum_{s \in S} x_s = 1$.
- *INA constraints:* Considering the limited number and processing capacity of programmable switches in the cluster, similar to [5, 11], we assume that each gradient will be aggregated in-network once at most to balance the problem complexity and the network performance, which is $y_v^{s,w} \le z_v^s, \forall s, w, v$.
- *Routing constraints:* Each gradient must be routed from a worker to a PS for global aggregation through a feasible path. Specifically, if gradients from worker $w$ to PS $s$ are directly aggregated by PS without INA, we have $\sum_{p \in P_{s,w}} q_p^{s,w} = y^{s,w}, \forall s, w$. If gradients from worker $w$ to PS $s$ are aggregated on the programmable switch $v$, we have $\sum_{p \in P_{v,w}} q_p^{v,w} = y_v^{s,w}, \forall s, w, v$, and $\sum_{p \in P_{s,v}} q_p^{s,v} = y_v^{s,w}, \forall s, w, v$.
- *Switch capacity constraints:* Each programmable switch can only aggregate gradients at a limited rate due to switch processing capacity limitations. Therefore, we have $\sum_{s \in S} f \cdot x_s \cdot \sum_{w \in W} y_v^{s,w} + c(v) \le C(v), \forall v$.
- *Link capacity constraints:* For each link $e$, its traffic load should not exceed its bandwidth capacity $C(e)$. Thus, we have $\sum_{s \in S} f \cdot x_s \cdot \sum_{v \in V} \sum_{p \in P : e \in p} \left( q_p^{s,v} + \sum_{w \in W} (q_p^{v,w} + q_p^{s,w}) \right) + b(e) \le B(e), \forall e$.
- *PS capacity constraints:* For each PS $s$, the forwarding rate can't exceed its ingress bandwidth $B(s)$. For convenience, let binary variable $z_v^s$ indicates whether aggregated gradients exist on switch $v$ that need to be sent to PS $s$. Obviously, we have $y_v^{s,w} \le z_v^s, \forall s, w, v$. Note that two types of gradients are routed to the PS for global aggregation: gradients forwarded directly by workers without network aggregation (*i.e.*, $y^{s,w} = 1$), and gradients aggregated by programmable switches (*i.e.*, $z_v^s = 1$). Accordingly, we have $f \cdot x_s \cdot \left( \sum_{w \in W} y^{s,w} + \sum_{v \in V} z_v^s \right) + b(s) \le B(s), \forall s$.

Furthermore, We adopt a synchronous approach [28] for model updating, wherein the parameter servers (PSs) aggregate gradients after receiving them from all required workers. This method ensures system stability. In this context, a faster gradient sending rate generally leads to shorter communication times. To capture this, we introduce variable $f$ to represent the gradient sending rate of workers, with our objective being to maximize $f$. Formally, we define the problem as Eq. (1). The first equality in Eq. (1) represents the model partition constraints. The subsequent set of equalities denotes the INA constraints. Following that, the third to fifth sets

of equalities describe the routing constraints. The sixth set of inequalities represents the switch capacity constraints. The seventh set of inequalities represents the link capacity constraints. Finally, the last two inequalities denote the PS capacity constraints.

$$\max f$$

$$S.t. \begin{cases} \sum_{s \in S} x_s = 1, \\ y^{s,w} + \sum_{v \in V} y_v^{s,w} = 1, & \forall s, w \\ \sum_{p \in P_{s,w}} q_p^{s,w} = y^{s,w}, & \forall s, w \\ \sum_{p \in P_{v,w}} q_p^{v,w} = y_v^{s,w}, & \forall s, w, v \\ \sum_{p \in P_{s,v}} q_p^{s,v} = y_v^{s,w}, & \forall s, w, v \\ \sum_{s \in S} f \cdot x_s \cdot \sum_{w \in W} y_v^{s,w} + c(v) \leq C(v), & \forall v \\ \sum_{s \in S} f x_s \sum_{v \in V} \sum_{p \in P: e \in p} ((q_p^{s,v} + \sum_{w \in W} (q_p^{v,w} + q_p^{s,w})) + b(e) \leq B(e), & \forall e \\ y_v^{s,w} \leq z_v^s, & \forall s, w, v \\ f \cdot x_s \cdot (\sum_{w \in W} y^{s,w} + \sum_{v \in V} z_v^s) + b(s) \leq B(s), & \forall s \\ x_s \in [0, 1], & \forall s \\ y^{s,w}, y_v^{s,w} \in \{0, 1\}, & \forall s, w, v \\ z_v^s \in \{0, 1\}, & \forall s, v \\ q_p^{s,w}, q_p^{v,w}, q_p^{s,v} \in \{0, 1\}, & \forall s, w, v, p \\ f \geq 0 \end{cases} \quad (1)$$

In fact, it is difficult to directly solve the problem in Eq. (1). Note that the left side of the sixth set of inequalities in Eq. (1) contains the product of two continuous variables $x_s$, $f$ and a binary variable $y_v^{s,w}$. In other words, InArt is a typically nonlinear mixed-integer programming (NMIP) problem, which is NP-hard [29]. Designing an algorithm for InArt is far from trivial and in urgent need.

## 4 ALGORITHM DESIGN

### 4.1 Algorithm Workflow

In a cluster, where multiple DT jobs or applications are running simultaneously, the network traffic can undergo significant changes. Thus, in order to adapt to traffic uncertainty/dynamics, we should update routing paths and INA policy frequently. However, modifying the model scale on the PSs during training is not feasible due to consistency concerns.

To address this challenge, we propose a two-phase approach to solve the InArt problem. In the first phase, conducted at longer intervals such as several hours or a day, we divide the model among multiple PSs without considering route selection (§4.2). This simplifies InArt into a nonlinear programming problem, which we solve using the Lagrange multiplier method. In the second phase, triggered by events such as network congestion, we focus on the selection of routing paths for INA (§4.3). We maintain a fixed model partition ratio and transform InArt into an integer programming problem. To efficiently handle this, we design a randomized rounding-based algorithm for INA with route selection.

### 4.2 Algorithm Design for Splitting the Model

In the first phase, we mainly split the model among multiple PSs and determine the sub-model that each PS is responsible for, $i.e.$, get the value of variables $x_s \in [0, 1]$. The procedure for this task is outlined in Alg. 1. At first, we focus on the switch capacity constraints and

the PS capacity constraints in Eq. (1). Then, we relax the variables $y^{s,w}, y_v^{s,w}$ and $z_v^s$ from integer to fractional. The problem in Eq. (1) converts to a nonlinear programming as follows:

$$\max f$$

$$S.t. \begin{cases} \sum_{s \in S} x_s = 1, \\ y^{s,w} + \sum_{v \in V} y_v^{s,w} = 1, & \forall s, w \\ \sum_{s \in S} f \cdot x_s \cdot \sum_{w \in W} y_v^{s,w} \leq C(v) - c(v), & \forall v \\ y_v^{s,w} \leq z_v^s, & \forall s, w, v \\ f \cdot x_s \cdot (\sum_{w \in W} y^{s,w} + \sum_{v \in V} z_v^s) \leq B(s) - b(s), & \forall s \\ x_s \in [0, 1], & \forall s \\ y^{s,w}, y_v^{s,w} \in [0, 1], & \forall s, w, v \\ z_v^s \in [0, 1], & \forall s, v \\ f \geq 0 \end{cases} \quad (2)$$

Note that variables $y^{s,w}$, $y_v^{s,w}$ and $z_v^s$ are integral in Eq. (1), but fractional in Eq. (2). Since Eq. (2) is a nonlinear programming, we design a generalized Lagrange multiplier method [30], called L-InArt, to get the value of variables $x_s$. Let symbol $X$ represents all variables in Eq. (2), $i.e.$, $X = \{x_s, y^{s,w}, y_v^{s,w}, z_v^s\}$. We consider the Lagrange function $\mathcal{L}(X)$ of Eq. (2) as follows:

$$\mathcal{L}(X) = \sum_{s \in S} \sum_{w \in W} \lambda_{s,w} h_{s,w}(X) + \alpha w(X) + \sum_{v \in V} \rho_v p_v(X)$$
$$+ \sum_{s \in S} \theta_s q_s(X) + \sum_{s \in S} \sum_{w \in W} \sum_{v \in V} \sigma_{v,s,w} r_{v,s,w}(X)$$
$$- f - \tau_s x_s - \beta_{s,w} y_{s,w} - \delta_{s,v,w} y_{s,v,w} - \zeta_{s,v} z_{s,v} + \eta_s (x_s - 1)$$
$$+ \mu_{s,w}(y_{s,w} - 1) + \omega_{s,v,w}(y_{s,v,w} - 1) + \gamma_{s,v}(z_{s,v} - 1) \quad (3)$$

Greek variables in Eq. (3) represent the Lagrange multiplier corresponding to the constraints in Eq. (2). For example, the variable $\alpha$ corresponds to the first set of constraints of $x_s$ in Eq. (2). Meanwhile, these variables should be non-negative. In addition, the functions $h_{s,w}(X)$, $w(X)$, $p_v(X)$, $q_s(X)$, and $r_{v,s,w}(X)$ denote the first set to the fifth set of constraints in Eq. (2), respectively. The definition of these functions is as follows:

$$\begin{cases} h_{s,w}(X) = y^{s,w} + \sum_{v \in V} y_v^{s,w} - 1, & \forall s, w \\ w(X) = 1 - \sum_{s \in S} x_s, \\ p_v(X) = \sum_{s \in S} f \cdot x_s \cdot \sum_{w \in W} y_v^{s,w} - (C(v) - c(v)) & \forall v \\ q_s(X) = f \cdot x_s \cdot (\sum_{w \in W} y^{s,w} + \sum_{v \in V} z_v^s) - (B(s) - b(s)), & \forall s \\ r_{v,s,w}(X) = y_v^{s,w} - z_v^s, & \forall v, s, w \end{cases} \quad (4)$$

To determine the extreme point, we utilize the Karush-Kuhn-Tucker (KKT) conditions [30, 31] and find the partial derivative, which yields a set of equations for $x_s$. Solving these equations through Gaussian elimination [32] provides us with the values of $x_s$. Due to space limit, the reader can refer to [30, 33] for a more comprehensive understanding. Once we have obtained the calculated values of $x_s$, we split the DT model among multiple servers accordingly.

### 4.3 Algorithm Design for INA and Routing

The second phase of InArt gives the INA and routing schemes. Since the value of $x_s$ is solved in the first phase, we introduce the

---

**Algorithm 1:** L-InArt: Lagrange Multiplier Algorithm for InArt

---

**1 Step 1: Relaxing the InArt problem**
**2** Focus on the switch capacity constraints and the PS capacity constraints in Eq. (1)
**3** Relax the variables $y^{s,w}$, $y_v^{s,w}$, and $z_v^s$ from integer to fractional
**4** Construct a nonlinear programming in Eq. (2)
**5 Step 2: Deriving the extreme point of $x_s$**
**6** Give the Lagrange function $\mathcal{L}(X)$ in Eq. (3)
**7** Obtain an equation set for $x_s$ by take partial derivative to the Lagrange function $\mathcal{L}(X)$ of Eq. (3) and Eq. (4)
**8** Solve these equations and split the model among multiple PSs based on the value of $x_s$

---

result into Eq. (1), and simplify InArt into an integer linear programming problem, which is challenging to solve in a polynomial time. Accordingly, in this section, we propose a randomized rounding-based algorithm for the second phase, called R-InArt. The R-InArt algorithm is formally described in Alg. 2.

In the first step of R-InArt, we construct linear programming as relaxation of Eq. (1). Specifically, InArt assumes that each gradient will be routed on a feasible path and aggregated on at most one switch. By relaxing these assumptions, each gradient is splittable, can be routed through several feasible paths and aggregated by multiple switches. We formulate the linear programming LP-InArt as follows:

$$\max f$$

$$S.t.\begin{cases} y^{s,w} + \sum_{v \in V} y_v^{s,w} = 1, & \forall s, w \\ \sum_{p \in P_{s,w}} q_p^{s,w} = y^{s,w}, & \forall s, w \\ \sum_{p \in P_{v,w}} q_p^{v,w} = y_v^{s,w}, & \forall s, w, v \\ \sum_{p \in P_{s,v}} q_p^{s,v} = y_v^{s,w}, & \forall s, w, v \\ \sum_{s \in S} f \cdot x_s \cdot \sum_{v \in V} y_v^{s,w} + c(v) \le C(v), & \forall v \\ \sum_{s \in S} f x_s \sum_{v \in V} \sum_{p \in P: e \in p} ((q_p^{s,v} + \sum_{w \in W}(q_p^{v,w} + q_p^{s,w})) + b(e) \le B(e), & \forall e \\ y_v^{s,w} \le z_v^s, & \forall s, w, v \\ f \cdot x_s \cdot (\sum_{w \in W} y^{s,w} + \sum_{v \in V} z_v^s) + b(s) \le B(s), & \forall s \\ y^{s,w}, y_v^{s,w} \in [0,1], & \forall s, w, v \\ z_v^s \in [0,1], & \forall s, v \\ q_p^{s,w}, q_p^{v,w}, q_p^{s,v} \in [0,1], & \forall s, w, v, p \\ f \ge 0 \end{cases}$$ (5)

Note that variables $y^{s,w}$, $y_v^{s,w}$, $z_v^s$, $q_p^{s,w}$, $q_p^{v,w}$, and $q_p^{s,v}$ are integer in Eq. (1), but fractional in Eq. (5). Since Eq. (5) is a linear programming problem, we can use a linear programming solver (*e.g.*, Cplex [34]) to solve it in polynomial time. Assume that the optimal solution for Eq. (5) is denoted as $\{\widetilde{y}^{s,w}, \widetilde{y}_v^{s,w}, \widetilde{z}^{s,v}, \widetilde{q}_p^{s,w}, \widetilde{q}_p^{v,w}, \widetilde{q}_p^{s,v}\}$, and the optimal result is denoted as $\widetilde{f}$. Since Eq. (5) is a relaxation of Eq. (1), $\widetilde{f}$ is the upper-bound for Eq. (1).

In the second step of R-InArt, we give the INA scheme and routing Path. At first, using the randomized rounding (RR) method [35], we derive the integral solution $\{\widehat{y}^{s,w}, \widehat{y}_v^{s,w}\}$, for $\forall s \in S$, $\forall w \in W$, and $v \in V$. Specifically, if $\widehat{y}^{s,w} = 1$, it means that gradients from

---

**Algorithm 2:** R-InArt: RR-Based Algorithm for InArt

---

**1 Step 1: Solving the relaxed problem of Eq.** (1)
**2** Construct the linear programming LP-InArt in Eq. (5)
**3** Derive the optimal solution $\widetilde{y}^{s,w}$, $\widetilde{y}_v^{s,w}$, $\widetilde{z}^{s,v}$, $\widetilde{q}_p^{s,w}$, $\widetilde{q}_p^{v,w}$, and $\widetilde{q}_p^{s,v}$
**4 Step 2: Selecting Routing Path**
**5** Obtain an integer solution $\widehat{y}^{s,w}$ and $\widehat{y}_v^{s,w}$ by RR
**6 for** *each PS $s \in S$* **do**
**7**   **for** *each worker $w \in W$* **do**
**8**     **if** *$\widehat{y}^{s,w}$ == 1* **then**
**9**       Obtain an integral solution $\widehat{q}_p^{s,w}$ by RR
**10**       **for** *each path $p \in P_{s,w}$* **do**
**11**         **if** *$\widehat{q}_p^{s,w}$ == 1* **then**
**12**           **for** *each switch $v$ along path $p$* **do**
**13**             Install a flow entry on switch $v$
**14**     **for** *each switch $v \in V$* **do**
**15**       **if** *$\widehat{y}_v^{s,w}$ == 1* **then**
**16**         Set the value of $z_v^s$ to 1
**17**         Install a INA rule on switch $v$ for the flow from worker $w$ to PS $s$
**18**         Obtain an integral solution $\widehat{q}_p^{v,w}$ by RR
**19**         **for** *each path $p \in P_{v,w}$* **do**
**20**           **if** *$\widehat{q}_p^{v,w}$ == 1* **then**
**21**             **for** *each $v$ along path $p$* **do**
**22**               Install a flow entry on $v$
**23**     **for** *each switch $v \in V$* **do**
**24**       Obtain an integral solution $\widehat{q}_p^{s,v}$ by RR
**25**       **for** *each path $p \in P_{s,v}$* **do**
**26**         **if** *$\widehat{q}_p^{s,v}$ == 1* **then**
**27**           **for** *each switch $v$ along path $p$* **do**
**28**             Install a flow entry on switch $v$

---

worker $w$ to PS $s$ will not be aggregated by any switches, but will be aggregated on PS $s$. If $\widehat{y}_v^{s,w} = 1$, it means that the gradient from worker $w$ to PS $s$ will be aggregated on switch $v$. Moreover, we will set the value of $z_v^s$ as 1. Next, we give the routing path $q$ of each gradient from worker $w$ to PS $s$. Then we derive the integral solution by RR, denoted as $\{\widehat{q}_p^{s,w}, \widehat{q}_p^{v,w}, \widehat{q}_p^{s,v}\}$.

Note that each gradient will be aggregated in at most one switch for INA, and be assigned one feasible path for routing by InArt. In the following, we take the rounding process of INA as an example to illustrate the specific RR details of the R-InArt algorithm. Specifically, there are two switches for INA, and the optimal solution $\widehat{y}^{s,w}$ and $\{\widehat{y}_v^{s,w}\}$ of a worker $w$ equals to 0.1 and $\{0.4, 0.5\}$, respectively. Then the interval $[0, 1]$ is splitted into three parts: $(0, 0.4]$, $(0.4, 0.9]$, and $(0.9, 1]$. We generate a random value between 0 to 1, and choose at most one switch for INA depending on this value. If the value is less than 0.4, R-InArt will choose the first switch as the aggregation switch for the gradient from worker $w$ to PS $s$. Otherwise, if the value is larger than 0.4 and less than 0.9, then the controller will choose the second switch as the aggregation switch for this gradient. Meanwhile, if the value is larger than 0.9, the gradient will be aggregated on parameter servers but not aggregated in the cluster.

**Approximation Performance:** The approximate factors of our algorithm are bi-criteria approximation with respect to both

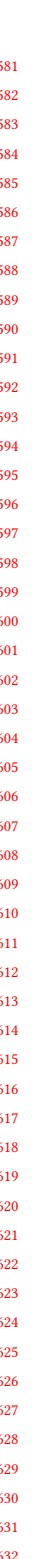

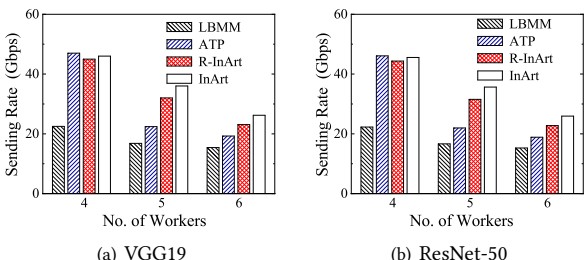

(a) VGG19       (b) ResNet-50

**Figure 2: Gradient Sending Rate vs. No. of Workers**

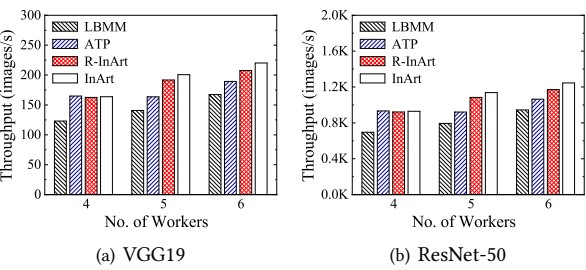

(a) VGG19       (b) ResNet-50

**Figure 3: Training Throughput vs. No. of Workers**

the objective value and resource constraints. Due to space limit, we omit the proofs of the approximation performance analysis of R-InArt. The reader can refer to [36–39] for the performance analysis of the randomized rounding method (RR). The approximation factor is $O(2\log|V|/\varphi + 2, 2\log|S|/\varphi + 2, 2\log|E|/\varphi + 2)$, which represents the approximation factors of switch capacity constraints, PS capacity constraints and link capacity constraints, respectively. Here $\varphi = \min\{\frac{2\cdot(C(v)-c(v))}{x_s\cdot f}, \frac{2\cdot(B(e)-b(e))}{x_s\cdot f}, \frac{2\cdot(B(s)-b(s))}{x_s\cdot f}, \forall v, e, s\}$ is a constant value about system. In practical scenarios, these factors are constant. We estimate $\varphi$ as 10, then the approximation factor becomes 2.78, 2.73, and 2.27, respectively.

## 5 EVALUATION

### 5.1 Performance Metrics and Benchmarks

*5.1.1 Performance Metrics.* We adopt the following eight performance metrics to evaluate the improvement of our proposed InArt for DT: (1) the gradient sending rate of workers; (2) the training throughput; (3) the per-iteration time; (4) the communication time; (5) the training speed; (6) the accuracy over training time; (7) the network throughput; (8) the ingress traffic amount of PSs.

During a testbed run, we use iftop [40] to monitor the egress bandwidth of each worker as *the gradient sending rate.* We measure the number of processed samples (*e.g.*, images) per second as *the training throughput.* In addition, we record the time between two consecutive iterations as *the per-iteration time.* In each iteration, we measure the duration from a worker sending gradients to receiving the updated model as *the communication time* of one iteration. Furthermore, we record the number of iterations over a period of time as *the training speed* and record *the accuracy of each iteration.*

During an emulation run, we measure *the gradient sending rate* and *the communication time.* In each iteration of the emulation experiment, we calculate the traffic volume of gradient transferred by all the links as *the network throughput.* In addition, we measure the total traffic volume of gradients from the workers and programmable switches to PSs per iteration, as *the ingress traffic amount of PSs.*

*5.1.2 Benchmarks.* We choose three benchmarks for performance comparison. The first benchmark splits the model in the same proportion among multiple PSs (*e.g.*, a DT architecture contains four PSs, each maintaining 25% of the total model), and then performs R-InArt for gradient route selection. The second benchmark is the Load Balance Min-Min scheduling (LBMM) algorithm [13]. LBMM is an efficient routing algorithm that without consider INA in the

cluster. For gradients from workers to PSs, LBMM chooses the routing path with the most negligible impact on routing load balancing. The third benchmark, called ATP [5], is a state-of-the-art INA method. In ATP, gradients are aggregated on ToR programmable switches in the cluster. Then the aggregated traffic will be routed to the PSs from the ToR switches with the least link load. Note that ATP does not involve model splitting, and for the purpose of fair comparison, we assume that the model is split equally across PSs in the following evaluations.

### 5.2 Testbed Evaluation

*5.2.1 Testbed Settings.* We use eight servers running Ubuntu 18.04 (Linux kernel version 5.4) and two Wedge100BF-32x programmable switches with Intel Tofino chip [20] to build the testbed. The topology of the testbed is the same as that of the example (Fig. 1) in §2.1. Specifically, all servers have a 22-core Intel Xeon 6152 processor, 128GB RAM, and an NVIDIA GeForce RTX 3090. Each server is equipped with a Mellanox ConnectX-6 dual-port 100Gbps NIC. Besides, all the servers are connected with programmable switches via 100Gbps links.

In terms of implementation details, similar to [11], we run PyTorch on each worker to carry out DT jobs. To implement INA on the switch, we write the P4 program in P4-16 with Tofino Native Architecture (TNA) [41]. More specifically, we pre-calculate our solution's model splitting and routing scheme with Pyomo [42] and install the corresponding entries to the programmable switches using the Barefoot Runtime Interface (BRI). We train two popular models on the Cifar-100 dataset [43]: ResNet50 [44] with a size of 97MB and VGG19 [45] with a size of 548MB. Specifically, the Cifar-100 dataset contains 60000 images, 50000 for training and 10000 for testing, labeled in 100 classes. Besides, the batch size is set as 32 for all training jobs. We run each testbed 30 times and calculate the average value as the results.

*5.2.2 Testbed Results.* We run three sets of experiments to evaluate the performance of InArt and benchmarks. In the first set of experiments, we observe the gradient sending rate of workers and the training throughput, as shown in Figs. 2-3. It is evident that InArt can achieve the best performance among all solutions. Fig. 2 shows that as the number of workers increases, InArt always obtains the highest gradient sending rate. For example, given 6 workers in VGG19, the gradient sending rate of InArt, R-InArt, ATP and LBMM are 26.2Gbps, 23.1Gbps, 19.25Gbps and 15.4Gbps, respectively. It means that InArt can increase the gradient sending rates by 13.4%,

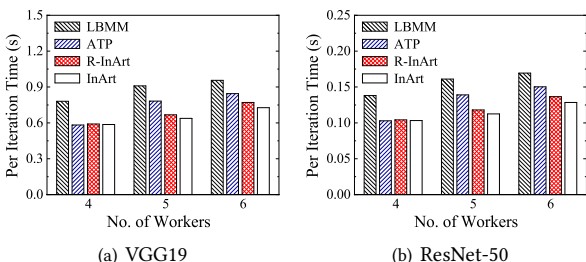

(a) VGG19        (b) ResNet-50

**Figure 4: Per Iteration Time vs. No. of Workers**

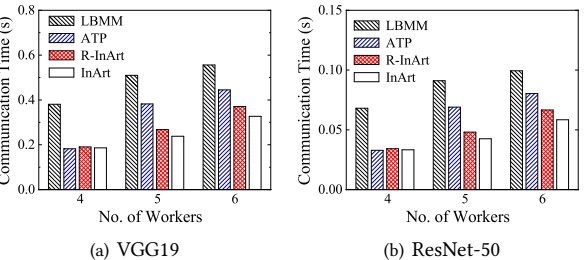

(a) VGG19        (b) ResNet-50

**Figure 5: Communication Time vs. No. of Workers**

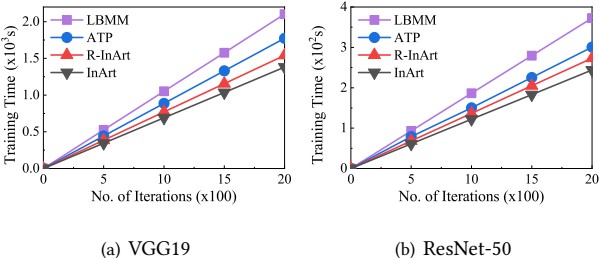

(a) VGG19        (b) ResNet-50

**Figure 6: Training Time vs. No. of Iterations**

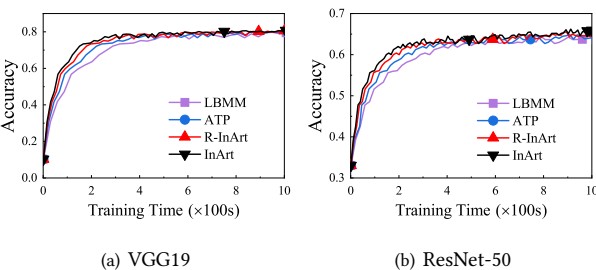

(a) VGG19        (b) ResNet-50

**Figure 7: Accuracy over Training Time**

36.5% and 72%, compared with R-InArt, ATP and LBMM, respectively. In Fig. 3, InArt consistently achieves the highest training throughput with increasing numbers of workers. Specifically, with 6 workers in VGG19, InArt achieves a throughput of 223 images/s. Comparatively, R-InArt, ATP, and LBMM achieve throughputs of 206 images/s, 188 images/s, and 162 images/s, respectively. In other words, InArt can improve the training throughput by 8.3%, 18.6% and 37.7% compared with R-InArt, ATP and LBMM, respectively. The reason is that InArt selects a proper routing path under the INA framework and designs an appropriate model splitting scheme to maximize the gradient sending rate of workers.

The second set of experiments measures the total time and the communication time of one iteration. Fig. 4 shows the per-iteration time with different numbers of workers. Note that per-iteration time consists of the local training time and the communication time. Our method doesn't optimize the local training time but can co-exist with solutions decreasing local training time if needed. We observe that the per-iteration time increases as the number of workers increases, while InArt always obtains the least per-iteration time. For example, when the number of workers is 6 in VGG19, the per-iteration times of LBMM, ATP, R-InArt and InArt are 0.97s, 0.85s, 0.78s and 0.69s, respectively. That means, InArt reduces the per-iteration time by 29%, 19% and 12% compared with LBMM, ATP and R-InArt, respectively. As shown in Fig. 5, InArt always has the shortest communication time in each iteration. Given 6 workers in VGG19, the communication time of LBMM, ATP, R-InArt and InArt are 0.56s, 0.45s, 0.38s and 0.31s, respectively. InArt decreases the communication time by 45%, 32% and 19%, compared with LBMM, ATP and R-InArt, respectively. The reason is that InArt has the highest gradient sending rate of workers (as described in Fig. 2), thereby reducing the communication time.

Finally, we run two DT jobs at 6 workers to evaluate the performance of training time and accuracy. From Figs. 6-7, we can

conclude that InArt always takes the least time to complete the same number of iterations compared with other alternatives. It can be observed from Fig. 6 that InArt takes the least time to complete the DT job. For example, it takes 1380s for InArt to complete 2000 iterations of the VGG19 training job, while the number are 1541s, 1775s and 2105s when we use R-InArt, ATP and LBMM, respectively. Fig. 7 shows that InArt can obtain the specified accuracy with the least time. For instance, when the mode is VGG19, InArt first achieves an accuracy of 0.7214 in 151s, while the time of R-InArt, ATP and LBMM are 181s, 208s and 278s, respectively. It means that InArt can reach the target accuracy 1.2×, 1.38× and 1.84× faster than R-InArt, ATP and LBMM, respectively. The results show that proper gradient routing with INA can significantly speed up the distributed model training.

### 5.3 Emulation Evaluation

*5.3.1 Emulation Settings.* We implement a middle-scale emulation with the classical fat-tree topology [46], which is commonly adopted in clusters. We use the mininet tool [47] to implement the fat-tree topology, which consists of 9 core switches, 18 aggregation switches, 18 ToR switches, and 54 servers. We randomly selected 4 servers as PSs and the remaining servers as workers. Since we cannot support P4 hardware switches of a certain scale, we obtain results using bmv2 [48] software switches. Unfortunately, bmv2 software switches are not designed for line-rate packet processing [49]. Therefore, we cannot inject Gbps traffic into bmv2 switches for our evaluations, and shrink the experimental setup by a factor of 1000.

These evaluations are performed under two common network scenarios. The first is a homogeneous scenario, in which the capacity of each link is 20Mbps. The second is a heterogeneous scenario, and the link capacity is randomly generated between 10Mbps and 30Mbps. We set the processing capacity of PSs and aggregation capacity of bmv2 switches as 20Mbps and 9Mbps, respectively. To

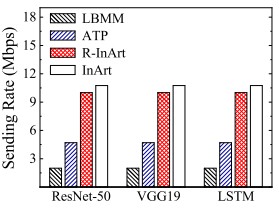
(a) Homogeneous Network

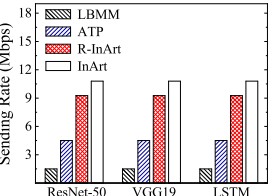
(b) Heterogeneous Network

Figure 8: Gradient Sending Rate in Different Models

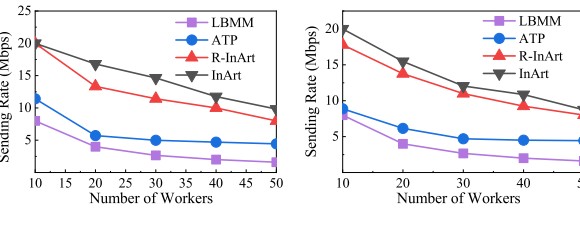
(a) Homogeneous Network   (b) Heterogeneous Network

Figure 9: Gradient Sending Rate vs. No. of Workers

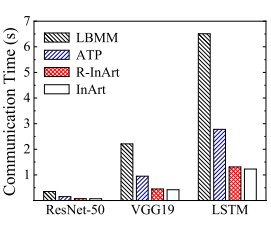
(a) Homogeneous Network

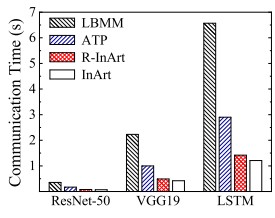
(b) Heterogeneous Network

Figure 10: Gradient Communication Time in Different Models

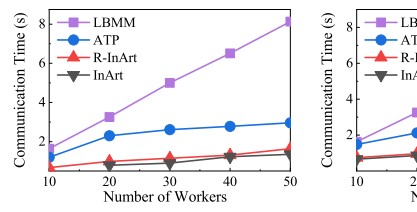
(a) Homogeneous Network   (b) Heterogeneous Network

Figure 11: Gradient Communication Time vs. No. of Workers

implement the INA and routing, we pre-program the INA logic of bmv2 switches and pre-install the flow table by P4 language. The emulation tests three different models, LSTM, VGG19, and ResNet-50. Specifically, the LSTM model is commonly used for time series prediction, and VGG19 and ResNet-50 are widely used for image classification. We set the gradient size in one iteration of LSTM, VGG19 and ResNet-50 by a factor of 1000 to 1627KB, 548KB, and 97KB, respectively [5]. To emulate the synchronous gradient communications, we deploy tcpreplay [50] tools on each worker to send packets at the same rate. We run each emulation 30 times and calculate the average value as the results.

*5.3.2 Emulation Results.* We run three sets of experiments for performance evaluations. The first set of experiments compares the sending rate of workers, as shown in Figs. 8-9. From the left plot of Fig. 8, when training LSTM jobs, the sending rate of workers is 10.75Mbps and 4.7Mbps by InArt and ATP, respectively. In Fig. 9, as the number of workers increases, the gradient sending rate will gradually decrease since more workers can use more network resources in the cluster. Inspiringly, our solution can achieve a faster sending rate than other benchmarks. From the right plot of Fig. 9, when there are 50 workers in the network, the sending rate of workers is 8.73Mbps and 4.44Mbps by InArt and ATP, respectively. Based on the evaluation results, our solution improved the sending rate by 97% compared with ATP. That is because while we perform traffic aggregation operations on edge switches, some core switches, and aggregation switches may also participate in INA.

The second set of experiments observes the communication time of one iteration. In Fig. 10, we first observe the gradient communication time of three models. As the gradient size increases, the communication time will become longer. Obviously, the communication time by InArt is much slower than that of ATP and LBMM. From the left plot in Fig. 10, in the homogeneous scenario, the gradient communication time of Resnet-50 is 0.068s, 0.152s, and

0.35s by InArt, ATP, and LBMM, respectively. Similarly, our solution performs better than other benchmarks in the heterogeneous scenario. In Fig. 11, we observe the impact of the number of workers on communication time. As the number of workers increases, the communication time accordingly increases. However, the increasing rate of InArt is much slower than that of ATP and LBMM. For example, when there are 50 workers in the right plot of Fig. 11, the communication time are 1.52s, 8.13s and 2.96s corresponding to our solution, LBMM and ATP, respectively. That means our solution reduces the communication time by 81% and 49% compared with LBMM and ATP, respectively. That is because a faster sending rate can effectively reduce communication time.

Our third set of experiments measures the network throughput and the ingress traffic amount of PSs per iteration. Due to space constraints, we present a summary of the results, while more detailed results can be found in §A. InArt significantly improves the network throughput compared to state-of-the-art INA works, achieving approximately 1.6× higher throughput. Additionally, our approach reduces the load on PSs by 53%. These improvements are attributed to InArt's utilization of a combined INA scheme that incorporates submodel partitioning and route selection.

## 6 CONCLUSION

In this paper, we design and implement InArt, the first-of-its-kind work on INA with route selection in a multi-PS architecture, to accelerate distributed training. InArt utilizes a multi-PS architecture to distribute DT tasks among multiple PSs and effectively selects routing schemes to fully leverage the capabilities of INA. Due to traffic dynamics, InArt takes a two-phase approach: splitting the training model among multiple PSs and selecting the routing paths for INA. Two algorithms have been designed for these two phases, respectively. Experiment results show that InArt can achieve a superior gradient sending rate and less communication time than the state-of-the-art solutions.

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

# A ADDITIONAL EVALUATION DETAILS

## A.1 Network Throughput

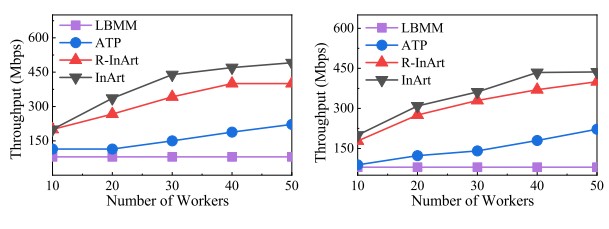

(a) Homogeneous Network    (b) Heterogeneous Network

**Figure 12: Network Throughput vs. No. of Workers**

As shown in Fig. 12, the network throughput gradually increases with the increasing number of workers, while our solution has the highest throughput. For example, when there are 30 workers in the left plot of Fig. 12, InArt, ATP, and LBMM achieves a network throughput of 420, 160, and 90 Mbps, respectively. This suggests that our INA and dynamic routing approach utilizes network resources more efficiently. Compared with ATP, InArt improves the network throughput by about 1.6×.

## A.2 Ingress Traffic amount of PS

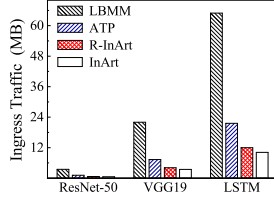 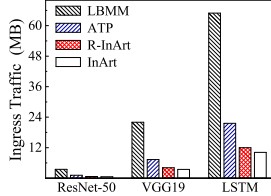

(a) Homogeneous Network    (b) Heterogeneous Network

**Figure 13: Ingress Traffic amount of PSs in Different Models.**

In Fig. 13, we indicate the ingress traffic amount of PSs. As expected that the ingress traffic amount of PSs using LBMM is much higher than that of other benchmarks. This is because the LBMM does not consider INA, and all the gradients will be aggregated on PSs. Note that InArt significantly reduces the processing load on PSs. For example, when training VGG19, the ingress traffic amount of PSs is 3.44MB, 7.33MB, and 22MB by InArt, ATP, and LBMM, respectively. Compared with ATP, our method reduces the load on PSs by 53%.

