# OpenReview forum: "InArt:  In-Network Aggregation with Route Selection  for Accelerating Distributed Training"
_ACM.org/TheWebConf/2024/Conference — TheWebConf24_

### Official Review · Reviewer_VuMr · 2023-11-17

**Novelty:** 4
**Technical Quality:** 4

**Review:**

This paper proposes a mechanism InArt to accelerate the distributed training process by In-network aggregation (INA) and path selection in the network. Implementation in P4 and evaluation show that InArt can reduce the communication time. Applying some computation ability in network equipment is a good way to accelerate large distribution computation task.

Strengths：

-A good try to use INA to improve the large distribution computation task.

Major weaknesses:

-The overhead and the loss of accuracy introduced by programmable switches have not been evaluated in the experiments.

**Questions:**

There are some limitations for this paper, and here are some comments which perhaps can be discussed further.

1)Model split is a key operation in InArt, however there are lacks of description how to split the model.

2)In the implementation, there are lack of P4 explanation and what kinds of work is conducted in the P4.

3)Analysis or evaluation for the loss of computation accuracy introduced by the INA can make this paper stronger.

4)There are many formulas in the paper. Some of them are not very useful and make this paper hard to follow.

**Reviewer Confidence:**

3: The reviewer is confident but not certain that the evaluation is correct

**Scope:**

2: The connection to the Web is incidental, e.g., use of Web data or API

---

### Official Review · Reviewer_HVFg · 2023-11-23

**Novelty:** 4
**Technical Quality:** 4

**Review:**

Summary Of Contributions:

The manuscript studies the problem of distributed training that use in network aggregation and multiple servers. This is an extension of previous work on distributed training, parameter servers, and in-network aggregation. All of these ideas were discussed in [5]. The main contribution is to build a linear programing problem that combines all the constraints into a single optimization problem. The solution is a two-phase approach of first dividing the model among multiples PS using the Lagrange multipliers and then using  a randomized rounding-based algorithm for INA with route selection between workers and servers.

The authors show in the evaluation of real and emulated switches that their solution performs better than other algorithms that did not consider the joint optimization problems.

The main new idea in the paper is the capability to split the model into independent sub-models that can reside independently in each of the servers. The joint optimization idea of splitting the model and routing the traffic is new. The solutions are mostly standard optimization approaches and are not very innovative. The main question is whether the joint optimization problem is realistic, especially if the arbitrary splits of the model into independent sub-models are at all feasible in real applications.

pros:

- The manuscript presents an extension to a known problem with practical network applications.

- The main contribution of the paper is being the first to apply INA with routing selection in a multi-PS architecture for accelerating distributed training.

- The introduction and motivation sections are well-written and easy to follow.

- The evaluation section is carried out extensively on both the hardware testbed and software emulation.

cons:

- The theoretical novelty of the manuscript is limited. The model and proposed solutions are mainly inherited from other works (model [5], optimization solutions [30, 33], and [35-39]) with limited new ideas.

- The manuscript claimed to “reduce communication time by 49% compared with state-of-art solutions” (lines 31-32). This statement is nit-picking as it is only true for one specific set of experimental results.

- The manuscript claimed that the proposed approach always achieved the best performances in testbed evaluation (line 690, 744). This is not true as it can be seen that for 4 workers, the proposed approach performed worse than other methods (i.e., ATP and R-InArt) in all metrics.

- The improvement of InArt over R-InArt was also marginal in most cases. Given that the main difference between these two is how the model is partitioned, this raises the question of the optimality of Algorithm 1.

- There is a lack of simulation studies of the algorithm’s performance when the number of PS varies.

- Other writing comments:

o The inequality equation for the INA constraints is incorrect (line 310).

o The notation “f” is used before formally introduced (line 321, 337).

o Minor typo errors (line 338, 519).

**Questions:**

- Can you please present a real use case of a machine learning model that can naturally be split into independent sub-models?
- How do you combine sub-models from multiple PSs and what is the cost associated with that?
- Can you explain why when there are 4 workers the InArt algorithm performs almost worse than others?
- Can you provide experiments with varying numbers of PSs?

**Reviewer Confidence:**

3: The reviewer is confident but not certain that the evaluation is correct

**Scope:**

3: The work is somewhat relevant to the Web and to the track, and is of narrow interest to a sub-community

---

### Official Review · Reviewer_jQFc · 2023-11-23

**Novelty:** 5
**Technical Quality:** 5

**Review:**

This paper designs an efficient route selection to accelerate distribution training (DT) by addressing the communication bottleneck at the parameter server (PS) in in-network aggregation (INA). The authors formulate a constrained programming problem that considers model splitting, dynamic traffic, and resource constraints. The optimization objective of this problem is set as the gradient sending rate, which could represent the efficiency of communication. To solve this nonlinear mixed-integer programming (NMIP) problem, the authors propose a two-phase approach. In the first phase, they develop a lagrange multiplier-based algorithm for model splitting in a multi-PS architecture.
In the second phase, the authors design a randomized rounding-based algorithm for INA and routing schemes. To validate the efficiency of the proposed method, the authors conduct extensive experiments on both hardware testbeds and software emulation environments. These experiments demonstrate the effectiveness of the proposed method in alleviating communication bottleneck and improving overall training efficiency.

**Strengths:**
* The paper is readable and well-organized.
* Detailed introduction and motivation explanation.
* Comprehensive experiments with clear conclusions.

**Weaknesses:**
* Explanations for some mathematical symbols were not provided in time (e.g., $z_v^s$ in Line 310 and $f$ in Line 320). Although the definitions were provided later, they still caused confusion on first reading.
* There is no further discussion of extending InArt to multi-DT job scenarios.

**Questions:**

1.Whether the first phase is to design a different model splitting scheme for each model, or to use the same scheme for all models?

2.If the latter, please explain whether it is reasonable to use the same model splitting scheme for different training tasks, considering the different model sizes of these tasks.

3.The paper states that the condition for triggering the execution of phase two is that congestion has occurred (in Line 395), please answer the following questions:
* For a single DT training job, how to measure the resource capacity $b(s),c(v)$ and $b(e)$ when congestion occurs during the training process. How to distinguish between resources occupied by background traffic and those occupied by the current task.
* Have there been any congestion issues when training the ResNet50, VGG16 and LSTM models? I would like the authors to discuss the scenario where congestion can be alleviated by rerouting paths when congestion occurs, thus speeding up the training of the model.

**Reviewer Confidence:**

3: The reviewer is confident but not certain that the evaluation is correct

**Scope:**

3: The work is somewhat relevant to the Web and to the track, and is of narrow interest to a sub-community

---

### Official Review · Reviewer_dMwe · 2023-11-27

**Novelty:** 6
**Technical Quality:** 7

**Review:**

This paper proposes InArt, a framework that allows in-network aggregation (INA) with routing selection in a multi-PS architecture by splitting distributed training (DT) tasks among multiple PSs. InArt is evaluated on both physical platforms and mininet emulation. Results show that communication time can be reduced by 49% compared to SoA.
Comments are provided per section.

Abstract: Provide the abbreviation for PS

Section 1 - Introduction
The flow of the introduction is very good. The first part motivates the use of Deep Learning (DL) in multiple domains and gives a brief overview of the concept of Distributed Training (DT). The second part provides a short description of DT's application in a PS architecture environment and introduces its performance bottlenect. A list of notable works are then presented that aim in solving the communication bottleneck issue in such systems using gradient compression and communication scheduling. However, particular limitations of these approaches motivate the use of in-network aggregation (INA) to address the communication bottleneck more efficiently. An overview of the paper's methodology along with its contributions are then presented.
While there is no related work section, I believe that the literature review is incorporated really well in the section.

Section 2 - Motivation:
This example provides a visual comparison between the proposed solution and two state of the art algorithms, namely LBMM and ATP. It shows that InArt can achieve the highest sending rate of 2.5 (while the other two solutions can achieve only 4/3 and 2, respectively). Figure 1 is clear and easy to understand.

Section 3 - Problem Definition
This section presents a detailed description of the system model architecture and formulates the problem. While all the parameters are explained thoroughly in 3.1, following equation 1 in 3.2 (constraints) is quite challenging for the reader (as the definition of each parameter is in the previous subsection)

Section 4 - Algorithm Design
This section presents the workflow of the algorithm for solving the InArt problem. The description of the algorithm is very detailed, however, similar to the previous section, it is hard to follow all the different algorithms and equations given the big number of involved parameters.

Section 5 - Evaluation
- The description of the performance metrics and the use of benchmarks for comparing the performance of the proposed solution is clear.
- Section 5.2.2: 'It is evident that InArt can achieve the best performance among all solutions'.
In subfigures 2a and 2b, InArt seems to have a slightly lower sending rate than ATP (No of Workers equals to 4), while, in subfigures 3a and 3b, the performance of InArt seems to be on par with ATP (No of Workers equals to 4).
- Overall the figures are very nice and align with the observations made in text.

Section 6 - Conclusions
Inclusion of future work would improve the paper.

**Questions:**

Q: It would be best to provide a table that outlines all available parameters along a short description that can be used as reference in the paper.
Q: Section 5.2.2. ' It means that InArt can increase the gradient sending rates by 13.4%, 36.5% and 72%, compared with R-InArt, ATP and LBMM, respectively.'
Is the increase here the average among the different counts of workers?
Similar question applies for the following 'InArt can improve the training throughput by 8.3%, 18.6% and 37.7% compared with R-InArt, ATP and LBMM, respectively.'

**Reviewer Confidence:**

3: The reviewer is confident but not certain that the evaluation is correct

**Scope:**

3: The work is somewhat relevant to the Web and to the track, and is of narrow interest to a sub-community

---

### Decision · Program_Chairs · 2024-01-22

**Decision:**

Accept

**Comment:**

The paper addresses a framework for in-network aggregation by efficient route selection.
 The reviewers mostly have high opinions about this paper, in terms of the novel use of INA with routing selection for accelerating distributed training, extensive evaluations, and clarity of the presentation, while the reviewer HVFg raised several nontrivial concerns -- calling for theoretical novelties. I also greatly appreciate the authors' effort in communicating with the reviewers, which brought resolutions to many concerns.

 I note that the rating distribution are mostly centered around `weak accept' but I do note the extensive discussion between the authors and the reviewers that brought multiple successful resolutions, I believe that this paper has been clarified much, and thus its merits outweigh its concerns. As the area chair, I'd like to recommend acceptance of this paper.